# Generalized Higher-Order Orthogonal Iteration for Tensor Decomposition and Completion

**Yuanyuan Liu**[†], **Fanhua Shang**[‡*], **Wei Fan**[§], **James Cheng**[‡], **Hong Cheng**[†]

[†]Dept. of Systems Engineering and Engineering Management,
The Chinese University of Hong Kong

[‡]Dept. of Computer Science and Engineering, The Chinese University of Hong Kong

[§]Huawei Noah′s Ark Lab, Hong Kong

{yyliu, hcheng}@se.cuhk.edu.hk {fhshang, jcheng}@cse.cuhk.edu.hk
david.fanwei@huawei.com

## Abstract

Low-rank tensor estimation has been frequently applied in many real-world problems. Despite successful applications, existing Schatten 1-norm minimization (SNM) methods may become very slow or even not applicable for large-scale problems. To address this difficulty, we therefore propose an efficient and scalable core tensor Schatten 1-norm minimization method for simultaneous tensor decomposition and completion, with a much lower computational complexity. We first induce the equivalence relation of Schatten 1-norm of a low-rank tensor and its core tensor. Then the Schatten 1-norm of the core tensor is used to replace that of the whole tensor, which leads to a much smaller-scale matrix SNM problem. Finally, an efficient algorithm with a rank-increasing scheme is developed to solve the proposed problem with a convergence guarantee. Extensive experimental results show that our method is usually more accurate than the state-of-the-art methods, and is orders of magnitude faster.

## 1 Introduction

There are numerous applications of higher-order tensors in machine learning [22, 29], signal processing [10, 9], computer vision [16, 17], data mining [1, 2], and numerical linear algebra [14, 21]. Especially with the rapid development of modern computing technology in recent years, tensors are becoming ubiquitous such as multi-channel images and videos, and have become increasingly popular [10]. Meanwhile, some values of their entries may be missing due to the problems in acquisition process, loss of information or costly experiments [1]. Low-rank tensor completion (LRTC) has been successfully applied to a wide range of real-world problems, such as visual data [16, 17], EEG data [9] and hyperspectral data analysis [9], and link prediction [29].

Recently, sparse vector recovery and low-rank matrix completion (LRMC) has been intensively studied [6, 5]. Especially, the convex relaxation (the Schatten 1-norm, also known as the trace norm or the nuclear norm [7]) has been used to approximate the rank of matrices and leads to a convex optimization problem. Compared with matrices, tensor can be used to express more complicated intrinsic structures of higher-order data. Liu et al. [16] indicated that LRTC methods utilize all information along each dimension, while LRMC methods only consider the constraints along two particular dimensions. As the generalization of LRMC, LRTC problems have drawn lots of attention from researchers in past several years [10]. To address the observed tensor with missing data, some weighted least-squares methods [1, 8] have been successfully applied to EEG data analysis, nature

---

[*]Corresponding author.

and hyperspectral images inpainting. However, they are usually sensitive to the given ranks due to their least-squares formulations [17].

Liu et al. [16] and Signorette et al. [23] first extended the Schatten 1-norm regularization for the estimation of partially observed low-rank tensors. In other words, the LRTC problem is converted into a convex combination of the Schatten 1-norm minimization (SNM) of the unfolding along each mode. Some similar algorithms can also be found in [17, 22, 25]. Besides these approaches described above, a number of variations [18] and alternatives [20, 28] have been discussed in the literature. In addition, there are some theoretical developments that guarantee the reconstruction of a low-rank tensor from partial measurements by solving the SNM problem under some reasonable conditions [24, 25, 11]. Although those SNM algorithms have been successfully applied in many real-world applications, them suffer from high computational cost of multiple SVDs as $O(NI^{N+1})$, where the assumed size of an $N$-th order tensor is $I \times I \times \cdots \times I$.

We focus on two major challenges faced by existing LRTC methods, the robustness of the given ranks and the computational efficiency. We propose an efficient and scalable core tensor Schatten 1-norm minimization method for simultaneous tensor decomposition and completion, which has a much lower computational complexity than existing SNM methods. In other words, our method only involves some much smaller unfoldings of the core tensor replacing that of the whole tensor. Moreover, we design a generalized Higher-order Orthogonal Iteration (gHOI) algorithm with a rank-increasing scheme to solve our model. Finally, we analyze the convergence of our algorithm and bound the gap between the resulting solution and the ground truth in terms of root mean square error.

## 2 Notations and Background

The mode-$n$ unfolding of an $N$th-order tensor $\mathcal{X} \in \mathbb{R}^{I_1 \times \cdots \times I_N}$ is a matrix denoted by $\mathcal{X}_{(n)} \in \mathbb{R}^{I_n \times \Pi_{j \neq n} I_j}$ that is obtained by arranging the mode-$n$ fibers to be the columns of $\mathcal{X}_{(n)}$. The Kronecker product of two matrices $A \in \mathbb{R}^{m \times n}$ and $B \in \mathbb{R}^{p \times q}$ is an $mp \times nq$ matrix given by $A \otimes B = [a_{ij}B]_{mp \times nq}$. The mode-$n$ product of a tensor $\mathcal{X} \in \mathbb{R}^{I_1 \times \cdots \times I_N}$ with a matrix $U \in \mathbb{R}^{J \times I_n}$ is defined as $(\mathcal{X} \times_n U)_{i_1 \cdots i_{n-1} j i_{n+1} \cdots i_N} = \sum_{i_n=1}^{I_n} x_{i_1 i_2 \cdots i_N} u_{j i_n}$.

### 2.1 Tensor Decompositions and Ranks

The CP decomposition approximates $\mathcal{X}$ by $\sum_{i=1}^{R} \mathbf{a}_i^1 \circ \mathbf{a}_i^2 \circ \cdots \circ \mathbf{a}_i^N$, where $R > 0$ is a given integer, $\mathbf{a}_i^n \in \mathbb{R}^{I_n}$, and $\circ$ denotes the outer product of vectors. The rank of $\mathcal{X}$ is defined as the smallest value of $R$ such that the approximation holds with equality. Computing the rank of the given tensor is NP-hard in general [13]. Fortunately, the $n$-rank of a tensor $\mathcal{X}$ is efficient to compute, and it consists of the matrix ranks of all mode unfoldings of the tensor. Given the $n$-rank$(\mathcal{X})$, the Tucker decomposition decomposes a tensor $\mathcal{X}$ into a core tensor multiplied by a factor matrix along each mode as follows: $\mathcal{X} = \mathcal{G} \times_1 U_1 \times_2 \cdots \times_N U_N$. Since the ranks $R_n$ $(n = 1, \cdots, N)$ are in general much smaller than $I_n$, the storage of the Tucker decomposition form can be significantly smaller than that of the original tensor. In [8], the weighted Tucker decomposition model for LRTC is

$$\min_{\mathcal{G}, \{U_n\}} \|\mathcal{W} \odot (\mathcal{T} - \mathcal{G} \times_1 U_1 \times_2 \cdots \times_N U_N)\|_F^2, \tag{1}$$

where the symbol $\odot$ denotes the Hadamard (elementwise) product, $\mathcal{W}$ is a nonnegative weight tensor with the same size as $\mathcal{T}$: $w_{i_1, i_2, \cdots, i_N} = 1$ if $(i_1, i_2, \cdots, i_N) \in \Omega$ and $w_{i_1, i_2, \cdots, i_N} = 0$ otherwise, and the elements of $\mathcal{T}$ in the set $\Omega$ are given while the remaining entries are missing.

### 2.2 Low-Rank Tensor Completion

For the LRTC problem, Liu et al. [16] and Signoretto et al. [23] proposed an extension of LRMC concept to tensor data as follows:

$$\min_{\mathcal{X}} \sum_{n=1}^{N} \alpha_n \|\mathcal{X}_{(n)}\|_*, \quad \text{s.t., } \mathcal{P}_\Omega(\mathcal{X}) = \mathcal{P}_\Omega(\mathcal{T}), \tag{2}$$

where $\|\mathcal{X}_{(n)}\|_*$ denotes the Schatten 1-norm of the unfolding $\mathcal{X}_{(n)}$, i.e., the sum of its singular values, $\alpha_n$'s are pre-specified weights, and $\mathcal{P}_\Omega$ keeps the entries in $\Omega$ and zeros out others. Gandy

et al. [9] presented an unweighted model, i.e., $\alpha_n = 1$, $n = 1, \ldots, N$. In addition, Tomioka and Suzuki [24] proposed a latent approach for LRTC problems:

$$\min_{\{\mathcal{X}_n\}} \sum_{n=1}^{N} \|(\mathcal{X}_n)_{(n)}\|_* + \frac{\lambda}{2}\|\mathcal{P}_\Omega(\sum_{n=1}^{N} \mathcal{X}_n) - \mathcal{P}_\Omega(\mathcal{T})\|_F^2. \tag{3}$$

In fact, each mode-$n$ unfolding $\mathcal{X}_{(n)}$ shares the same entries and cannot be optimized independently. Therefore, we need to apply variable splitting and introduce a separate variable to each unfolding of the tensor $\mathcal{X}$ or $\mathcal{X}_n$. However, all algorithms have to be solved iteratively and involve multiple SVDs of very large matrices in each iteration. Hence, they suffer from high computational cost and are even not applicable for large-scale problems.

## 3 Core Tensor Schatten 1-Norm Minimization

The existing SNM algorithms for solving the problems (2) and (3) suffer high computational cost, thus they have a bad scalability. Moreover, current tensor decomposition methods require explicit knowledge of the rank to gain a reliable performance. Motivated by these, we propose a scalable model and then achieve a smaller-scale matrix Schatten 1-norm minimization problem.

### 3.1 Formulation

**Definition 1.** *The Schatten 1-norm of an Nth-order tensor $\mathcal{X} \in \mathbb{R}^{I_1 \times \cdots \times I_N}$ is the sum of the Schatten 1-norms of its different unfoldings $\mathcal{X}_{(n)}$, i.e.,*

$$\|\mathcal{X}\|_* = \sum_{n=1}^{N} \|\mathcal{X}_{(n)}\|_*, \tag{4}$$

*where $\|\mathcal{X}_{(n)}\|_*$ denotes the Schatten 1-norm of the unfolding $\mathcal{X}_{(n)}$.*

For the imbalance LRTC problems, the Schatten 1-norm of the tensor can be incorporated by some pre-specified weights, $\alpha_n$, $n = 1, \ldots N$. Furthermore, we have the following theorem.

**Theorem 1.** *Let $\mathcal{X} \in \mathbb{R}^{I_1 \times \cdots \times I_N}$ with n-rank=$(R_1, \cdots, R_N)$ and $\mathcal{G} \in \mathbb{R}^{R_1 \times \cdots \times R_N}$ satisfy $\mathcal{X} = \mathcal{G} \times_1 U_1 \times_2 \cdots \times_N U_N$, and $U_n \in St(I_n, R_n)$, $n = 1, 2, \cdots, N$, then*

$$\|\mathcal{X}\|_* = \|\mathcal{G}\|_*, \tag{5}$$

*where $\|\mathcal{X}\|_*$ denotes the Schatten 1-norm of the tensor $\mathcal{X}$ and $St(I_n, R_n) = \{U \in \mathbb{R}^{I_n \times R_n} : U^T U = I_{R_n}\}$ denotes the Stiefel manifold.*

Please see Appendix A of the supplementary material for the detailed proof of the theorem. The core tensor $\mathcal{G}$ with size $(R_1, R_2, \cdots, R_N)$ has much smaller size than the observed tensor $\mathcal{T}$ (usually $R_n \ll I_n, n = 1, 2, \cdots, N$). According to Theorem 1, our Schatten 1-norm minimization problem is formulated into the following form:

$$\min_{\mathcal{G}, \{U_n\}, \mathcal{X}} \sum_{n=1}^{N} \|\mathcal{G}_{(n)}\|_* + \frac{\lambda}{2}\|\mathcal{X} - \mathcal{G} \times_1 U_1 \cdots \times_N U_N\|_F^2, \tag{6}$$

$$\text{s.t., } \mathcal{P}_\Omega(\mathcal{X}) = \mathcal{P}_\Omega(\mathcal{T}), \ U_n \in St(I_n, R_n), \ n = 1, \cdots, N.$$

Our tensor decomposition model (6) alleviates the SVD computation burden of much larger unfolded matrices in (2) and (3). Furthermore, we use the Schatten 1-norm regularization term in (6) to promote the robustness of the rank while the Tucker decomposition model (1) is usually sensitive to the given rank-$(r_1, r_2, \cdots, r_N)$ [17]. In addition, several works [12, 27] have provided some matrix rank estimation strategies to compute some values $(r_1, r_2, \cdots, r_N)$ for the $n$-rank of the involved tensor. In this paper, we only set some relatively large integers $(R_1, R_2, \cdots, R_N)$ such that $R_n \geq r_n$ for all $n = 1, \cdots, N$. Different from (2) and (3), some smaller matrices $V_n \in \mathbb{R}^{R_n \times \Pi_{j \neq n} R_j}$ $(n = 1, \cdots, N)$ are introduced into (6) as the auxiliary variables, and then our model (6) is reformulated into the following equivalent form:

$$\min_{\mathcal{G}, \{U_n\}, \{V_n\}, \mathcal{X}} \sum_{n=1}^{N} \|V_n\|_* + \frac{\lambda}{2}\|\mathcal{X} - \mathcal{G} \times_1 U_1 \cdots \times_N U_N\|_F^2, \tag{7}$$

$$\text{s.t., } \mathcal{P}_\Omega(\mathcal{X}) = \mathcal{P}_\Omega(\mathcal{T}), \ V_n = \mathcal{G}_{(n)}, \ U_n \in St(I_n, R_n), \ n = 1, \cdots, N.$$

In the following, we will propose an efficient gHOI algorithm based on alternating direction method of multipliers (ADMM) to solve the problem (7). ADMM decomposes a large problem into a series of smaller subproblems, and coordinates the solutions of subproblems to compute the optimal solution. In recent years, it has been shown in [3] that ADMM is very efficient for some convex or non-convex optimization problems in various applications.

## 3.2 A gHOI Algorithm with Rank-Increasing Scheme

The proposed problem (7) can be solved by ADMM. Its partial augmented Lagrangian function is

$$\mathcal{L}_\mu = \sum_{n=1}^{N}(\|V_n\|_* + \langle Y_n, \mathcal{G}_{(n)} - V_n\rangle + \frac{\mu}{2}\|\mathcal{G}_{(n)} - V_n\|_F^2) + \frac{\lambda}{2}\|\mathcal{X} - \mathcal{G} \times_1 U_1 \times_2 \cdots \times_N U_N\|_F^2, \quad (8)$$

where $Y_n$, $n = 1, \cdots, N$, are the matrices of Lagrange multipliers, and $\mu > 0$ is a penalty parameter. ADMM solves the proposed problem (7) by successively minimizing the Lagrange function $\mathcal{L}_\mu$ over $\{\mathcal{G}, U_1, \cdots, U_N, V_1, \cdots, V_N, \mathcal{X}\}$, and then updating $\{Y_1, \cdots, Y_N\}$.

**Updating $\{U_1^{k+1}, \cdots, U_N^{k+1}, \mathcal{G}^{k+1}\}$:** The optimization problem with respect to $\{U_1, \cdots, U_N\}$ and $\mathcal{G}$ is formulated as follows:

$$\min_{\mathcal{G}, \{U_n \in St(I_n, r_n)\}} \sum_{n=1}^{N} \frac{\mu^k}{2}\|\mathcal{G}_{(n)} - V_n^k + Y_n^k/\mu^k\|_F^2 + \frac{\lambda}{2}\|\mathcal{X}^k - \mathcal{G} \times_1 U_1 \cdots \times_N U_N\|_F^2, \quad (9)$$

where $r_n$ is an underestimated rank ($r_n \le R_n$), and is dynamically adjusted by using the following rank-increasing scheme. Different from HOOI in [14], we will propose a generalized higher-order orthogonal iteration scheme to solve the problem (9) in Section 3.3.

**Updating $\{V_1^{k+1}, \cdots, V_N^{k+1}\}$:** With keeping all the other variables fixed, $V_n^{k+1}$ is updated by solving the following problem:

$$\min_{V_n} \|V_n\|_* + \frac{\mu^k}{2}\|\mathcal{G}_{(n)}^{k+1} - V_n + Y_n^k/\mu^k\|_F^2. \quad (10)$$

For solving the problem (10), the spectral soft-thresholding operation [4] is considered as a shrinkage operation on the singular values and is defined as follows:

$$V_n^{k+1} = \text{prox}_{1/\mu^k}(M_n) := U\text{diag}(\max\{\sigma - \frac{1}{\mu^k}, 0\})V^T, \quad (11)$$

where $M_n = \mathcal{G}_{(n)}^{k+1} + Y_n^k/\mu^k$, $\max\{\cdot, \cdot\}$ should be understood element-wise, and $M_n = U\text{diag}(\sigma)V^T$ is the SVD of $M_n$. Here, only some matrices $M_n$ of smaller size in (11) need to perform SVD. Thus, this updating step has a significantly lower computational complexity $O(\sum_n R_n^2 \times \Pi_{j\neq n} R_j)$ at worst while the computational complexity of the convex SNM algorithms for both problems (2) and (3) is $O(\sum_n I_n^2 \times \Pi_{j\neq n} I_j)$ at each iteration.

**Updating $\mathcal{X}^{k+1}$:** The optimization problem with respect to $\mathcal{X}$ is formulated as follows:

$$\min_{\mathcal{X}} \|\mathcal{X} - \mathcal{G}^{k+1} \times_1 U_1^{k+1} \cdots \times_N U_N^{k+1}\|_F^2, \text{ s.t., } \mathcal{P}_\Omega(\mathcal{X}) = \mathcal{P}_\Omega(\mathcal{T}). \quad (12)$$

By deriving simply the KKT conditions for (12), the optimal solution $\mathcal{X}$ is given by

$$\mathcal{X}^{k+1} = \mathcal{P}_\Omega(\mathcal{T}) + \mathcal{P}_{\Omega^c}(\mathcal{G}^{k+1} \times_1 U_1^{k+1} \cdots \times_N U_N^{k+1}), \quad (13)$$

where $\Omega^c$ is the complement of $\Omega$, i.e., the set of indexes of the unobserved entries.

**Rank-increasing scheme:** The idea of interlacing fixed-rank optimization with adaptive rank-adjusting schemes has appeared recently in the particular context of matrix completion [27, 28]. It is here extended to our algorithm for solving the proposed problem. Let $\mathscr{U}^{k+1} = (U_1^{k+1}, U_2^{k+1}, \ldots, U_N^{k+1})$, $\mathscr{V}^{k+1} = (V_1^{k+1}, V_2^{k+1}, \ldots, V_N^{k+1})$, and $\mathscr{Y}^{k+1} = (Y_1^{k+1}, Y_2^{k+1}, \ldots, Y_N^{k+1})$. Considering the fact $\mathcal{L}_{\mu^k}(\mathcal{X}^{k+1}, \mathcal{G}^{k+1}, \mathscr{U}^{k+1}, \mathscr{V}^{k+1}, \mathscr{Y}^k) \le \mathcal{L}_{\mu^k}(\mathcal{X}^k, \mathcal{G}^k, \mathscr{U}^k, \mathscr{V}^k, \mathscr{Y}^k)$, our rank-increasing scheme starts $r_n$ such that $r_n \le R_n$. We increase $r_n$ to $\min(r_n + \triangle r_n, R_n)$ at iteration $k + 1$ if

$$\left|1 - \frac{\mathcal{L}_{\mu^k}(\mathcal{X}^{k+1}, \mathcal{G}^{k+1}, \mathscr{U}^{k+1}, \mathscr{V}^{k+1}, \mathscr{Y}^k)}{\mathcal{L}_{\mu^k}(\mathcal{X}^k, \mathcal{G}^k, \mathscr{U}^k, \mathscr{V}^k, \mathscr{Y}^k)}\right| \le \epsilon, \quad (14)$$

---

**Algorithm 1** Solving problem (7) via gHOI

**Input:** $\mathcal{P}_\Omega(\mathcal{T})$, $(R_1, \cdots, R_N)$, $\lambda$ and tol.
1: **while** not converged **do**
2:     Update $U_n^{k+1}$, $\mathcal{G}^{k+1}$, $V_n^{k+1}$ and $\mathcal{X}^{k+1}$ by (18), (20), (11) and (13), respectively.
3:     Apply the rank-increasing scheme.
4:     Update the multiplier $Y_n^{k+1}$ by $Y_n^{k+1} = Y_n^k + \mu^k(\mathcal{G}_{(n)}^{k+1} - V_n^{k+1})$, $n = 1, \ldots, N$.
5:     Update the parameter $\mu^{k+1}$ by $\mu^{k+1} = \min(\rho\mu^k, \mu_{\max})$.
6:     Check the convergence condition, $\max(\|\mathcal{G}_{(n)}^{k+1} - V_n^{k+1}\|_F^2, n = 1, \ldots, N) < $ tol.
7: **end while**
**Output:** $\mathcal{X}$, $\mathcal{G}$, and $U_n$, $n = 1, \cdots, N$.

---

which $\triangle r_n$ is a positive integer and $\epsilon$ is a small constant. Moreover, we augment $U_n^{k+1} \leftarrow [U_n^k, \widehat{U}_n]$ where $\widehat{H}_n$ has $\triangle r_n$ randomly generated columns, $\widehat{U}_n = (I - U_n^k(U_n^k)^T)\widehat{H}_n$, and then orthonormalize $\widehat{U}_n$. Let $\mathcal{V}_n = \text{refold}(V_n^k) \in \mathbb{R}^{r_1 \times \cdots \times r_N}$, and $\mathcal{W}_n \in \mathbb{R}^{(r_1 + \triangle r_1) \times \cdots \times (r_N + \triangle r_N)}$ be augmented as follows: $(\mathcal{W}_n)_{i_1, \cdots, i_N} = (\mathcal{V}_n)_{i_1, \cdots, i_N}$ for all $i_t \leq r_t$ and $t \in [1, N]$, and $(\mathcal{W}_n)_{i_1, \cdots, i_N} = 0$ otherwise, where refold$(\cdot)$ denotes the refolding of the matrix into a tensor and unfold$(\cdot)$ is the unfolding operator. Hence, we set $V_n^k = \text{unfold}(\mathcal{W}_n)$ and update $Y_n^k$ by the same way. We then update the involved variables $\mathcal{G}^{k+1}$, $V_n^{k+1}$ and $\mathcal{X}^{k+1}$ by (20), (11) and (13), respectively.

Summarizing the analysis above, we develop an efficient gHOI algorithm for solving the tensor decomposition and completion problem (7), as outlined in **Algorithm 1**. Our algorithm in essence is the Gauss-Seidel version of ADMM. The update strategy of Jacobi ADMM can easily be implemented, thus our gHOI algorithm is well suited for parallel and distributed computing and hence is particularly attractive for solving certain large-scale problems [21]. Algorithm 1 can be accelerated by adaptively changing $\mu$ as in [15].

### 3.3 Generalized Higher-Order Orthogonal Iteration

We propose a generalized HOOI scheme for solving the problem (9), where the conventional HOOI model in [14] can be seen as a special case of the problem (9) when $\mu^k = 0$. Therefore, we extend Theorem 4.2 in [14] to solve the problem (9) as follows.

**Theorem 2.** *Assume a real $N$th-order tensor $\mathcal{X}$, then the minimization of the following cost function*

$$f(\mathcal{G}, U_1, \ldots, U_N) = \sum_{n=1}^{N} \frac{\mu^k}{2} \|\mathcal{G}_{(n)} - V_n^k + Y_n^k/\mu^k\|_F^2 + \frac{\lambda}{2} \|\mathcal{X}^k - \mathcal{G} \times_1 U_1 \cdots \times_N U_N\|_F^2$$

*is equivalent to the maximization, over the matrices $U_1, U_2, \ldots, U_N$ having orthonormal columns, of the function*

$$g(U_1, U_2, \ldots, U_N) = \|\lambda \mathcal{M} + \mu^k \mathcal{N}\|_F^2, \tag{15}$$

*where $\mathcal{M} = \mathcal{X}^k \times_1 (U_1)^T \cdots \times_N (U_N)^T$ and $\mathcal{N} = \sum_{n=1}^{N} \text{refold}(V_n^k - Y_n^k/\mu^k)$.*

Please see Appendix B of the supplementary material for the detailed proof of the theorem.

**Updating $\{U_1^{k+1}, \cdots, U_N^{k+1}\}$:** According to Theorem 2, our generalized HOOI scheme successively solves $U_n$, $n = 1, \ldots, N$ with fixing other variables $U_j$, $j \neq n$. Imagine that the matrices $\{U_1, \ldots, U_{n-1}, U_{n+1}, \ldots, U_N\}$ are fixed and that the optimization problem (15) is thought of as a quadratic expression in the components of the matrix $U_n$ that is being optimized. Considering that the matrix has orthonormal columns, we have

$$\max_{U_n \in St(I_n, r_n)} \|\lambda \mathcal{M}_n \times_n U_n^T + \mu^k \mathcal{N}\|_F^2, \tag{16}$$

where

$$\mathcal{M}_n = \mathcal{X}^k \times_1 (U_1^{k+1})^T \cdots \times_{n-1} (U_{n-1}^{k+1})^T \times_{n+1} (U_{n+1}^k)^T \cdots \times_N (U_N^k)^T. \tag{17}$$

This is actually the well-known orthogonal procrustes problem [19], whose optimal solution is given by the singular value decomposition of $(\mathcal{M}_n)_{(n)}\mathcal{N}_{(n)}^T$, i.e.,

$$U_n^{k+1} = U^{(n)}(V^{(n)})^T, \tag{18}$$

where $U^{(n)}$ and $V^{(n)}$ are obtained by the skinny SVD of $(\mathcal{M}_n)_{(n)}\mathcal{N}_{(n)}^T$. Repeating the procedure above for different modes leads to an alternating orthogonal procrustes scheme for solving the maximization of the problem (16). For any estimate of those factor matrices $U_n$, $n = 1, \dots, N$, the optimal solution to the problem (9) with respect to $\mathcal{G}$ is updated in the following.

**Updating $\mathcal{G}^{k+1}$:** The optimization problem (9) with respect to $\mathcal{G}$ can be rewritten as follows:

$$\min_{\mathcal{G}} \sum_{n=1}^{N} \frac{\mu^k}{2} \|\mathcal{G}_{(n)} - V_n^k + Y_n^k/\mu^k\|_F^2 + \frac{\lambda}{2}\|\mathcal{X}^k - \mathcal{G} \times_1 U_1^{k+1} \cdots \times_N U_N^{k+1}\|_F^2. \quad (19)$$

(19) is a smooth convex optimization problem, thus we can obtain a closed-form solution,

$$\mathcal{G}^{k+1} = \frac{\lambda}{\lambda + N\mu^k} \mathcal{X}^k \times_1 (U_1^{k+1})^T \cdots \times_N (U_N^{k+1})^T + \frac{\mu^k}{\lambda + N\mu^k} \sum_{n=1}^{N} \text{refold}(V_n^k - Y_n^k/\mu^k). \quad (20)$$

# 4 Theoretical Analysis

In the following we first present the convergence analysis of Algorithm 1.

## 4.1 Convergence Analysis

**Theorem 3.** *Let $(\mathcal{G}^k, \{U_1^k, \dots, U_N^k\}, \{V_1^k, \dots, V_N^k\}, \mathcal{X}^k)$ be a sequence generated by Algorithm 1, then we have the following conclusions:*
*(I) $(\mathcal{G}^k, \{U_1^k, \dots, U_N^k\}, \{V_1^k, \dots, V_N^k\}, \mathcal{X}^k)$ are Cauchy sequences, respectively.*
*(II) If $\lim_{k\to\infty} \mu^k(V_n^{k+1} - V_n^k) = 0$, $n = 1, \cdots, N$, then $(\mathcal{G}^k, \{U_1^k, \dots, U_N^k\}, \mathcal{X}^k)$ converges to a KKT point of the problem (6).*

The proof of the theorem can be found in Appendix C of the supplementary material.

## 4.2 Recovery Guarantee

We will show that when sufficiently many entries are sampled, the KKT point of Algorithm 1 is stable, i.e., it recovers a tensor "close to" the ground-truth one. We assume that the observed tensor $\mathcal{T} \in \mathbb{R}^{I_1 \times I_2 \cdots \times I_N}$ can be decomposed as a true tensor $\mathcal{D}$ with rank-$(r_1, r_2, \dots, r_N)$ and a random gaussian noise $\mathcal{E}$ whose entries are independently drawn from $\mathcal{N}(0, \sigma^2)$, i.e., $\mathcal{T} = \mathcal{D} + \mathcal{E}$. For convenience, we suppose $I_1 = \cdots = I_N = I$ and $r_1 = \dots = r_N = r$. Let the recovered tensor $\mathcal{A} = \mathcal{G} \times_1 U_1 \times \dots \times_N U_N$, the root mean square error (RMSE) is a frequently used measure of the difference between the recovered tensor and the true one: $\text{RMSE} := \frac{1}{\sqrt{I^N}}\|\mathcal{D} - \mathcal{A}\|_F$.

[25] analyzes the statistical performance of the convex tensor Schatten 1-norm minimization problem with the general linear operator $\mathscr{X} : \mathbb{R}^{I_1 \times \dots \times I_N} \to \mathbb{R}^m$. However, our model (6) is non-convex for the LRTC problem with the operator $\mathcal{P}_\Omega$. Thus, we follow the sketch of the proof in [26] to analyze the statistical performance of our model (6).

**Definition 2.** *The operator $\mathcal{P}_S$ is defined as follows: $\mathcal{P}_S(\mathcal{X}) = P_{U_N} \cdots P_{U_1}(\mathcal{X})$, where $P_{U_n}(\mathcal{X}) = \mathcal{X} \times_n (U_n U_n^T)$.*

**Theorem 4.** *Let $(\mathcal{G}, U_1, U_2, \dots, U_N)$ be a KKT point of the problem (6) with given ranks $R_1 = \cdots = R_N = R$. Then there exists an absolute constant $C$ (please see Supplementary Material), such that with probability at least $1 - 2\exp(-I^{N-1})$,*

$$\text{RMSE} \le \frac{\|\mathcal{E}\|_F}{\sqrt{I^N}} + C\beta \left(\frac{I^{N-1}R\log(I^{N-1})}{|\Omega|}\right)^{\frac{1}{4}} + \frac{N\sqrt{R}}{C_1\lambda\sqrt{|\Omega|}}, \quad (21)$$

*where $\beta = \max_{i_1, \cdots, i_N} |\mathcal{T}_{i_1, \cdots, i_N}|$ and $C_1 = \frac{\|\mathcal{P}_S\mathcal{P}_\Omega(\mathcal{T}-\mathcal{A})\|_F}{\|\mathcal{P}_\Omega(\mathcal{T}-\mathcal{A})\|_F}$.*

The proof of the theorem and the analysis of lower-boundedness of $C_1$ can be found in Appendix D of the supplementary material. Furthermore, our result can also be extended to the general linear operator $\mathscr{X}$, e.g., the identity operator (i.e., tensor decomposition problems). Similar to [25], we assume that the operator satisfies the following restricted strong convexity (RSC) condition.

Table 1: RSE and running time (seconds) comparison on synthetic tensor data:

(a) Tensor size: $30 \times 30 \times 30 \times 30 \times 30$

| | WTucker | | WCP | | FaLRTC | | Latent | | gHOI | |
|---|---|---|---|---|---|---|---|---|---|---|
| SR | RSE±std. | Time | RSE±std. | Time | RSE±std. | Time | RSE±std. | Time | RSE±std. | Time |
| 10% | 0.4982±2.3e-2 | 2163.05 | 0.5003±3.6e-2 | 4359.23 | 0.6744±2.7e-2 | 1575.78 | 0.6268±5.0e-2 | 8324.17 | **0.2537±1.2e-2** | **159.43** |
| 30% | 0.1562±1.7e-2 | 2226.67 | 0.3364±2.3e-2 | 3949.57 | 0.3153±1.4e-2 | 1779.59 | 0.2443±1.2e-2 | 8043.83 | **0.1206±6.0e-3** | **143.86** |
| 50% | 0.0490±9.3e-3 | 2652.90 | 0.0769±5.0e-3 | 3260.86 | 0.0365±6.2e-4 | 2024.52 | 0.0559±7.7e-3 | 8263.24 | **0.0159±1.3e-3** | **135.60** |

(b) Tensor size: $60 \times 60 \times 60 \times 60$

| | WTucker | | WCP | | FaLRTC | | Latent | | gHOI | |
|---|---|---|---|---|---|---|---|---|---|---|
| SR | RSE±std. | Time | RSE±std. | Time | RSE±std. | Time | RSE±std. | Time | RSE±std. | Time |
| 10% | 0.2319±3.6e-2 | 1437.61 | 0.4766±9.4e-2 | 1586.92 | 0.4927±1.6e-2 | 562.15 | 0.5061±4.4e-2 | 5075.82 | **0.1674±3.4e-3** | **60.53** |
| 30% | 0.0143±2.8e-3 | 1756.95 | 0.1994±6.0e-3 | 1696.27 | 0.1694±2.5e-3 | 603.49 | 0.1872±7.5e-3 | 5559.17 | **0.0076±6.5e-4** | **57.19** |
| 50% | 0.0079±6.2e-4 | 2534.59 | 0.1335±4.9e-3 | 1871.38 | 0.0602±5.8e-4 | 655.69 | 0.0583±9.7e-4 | 6086.63 | **0.0030±1.7e-4** | **55.62** |

**Definition 3** (RSC). *We suppose that there is a positive constant $\kappa(\mathscr{X})$ such that the operator $\mathscr{X} : \mathbb{R}^{I_1 \times \cdots \times I_N} \to \mathbb{R}^m$ satisfies the inequality*

$$\frac{1}{m}\|\mathscr{X}(\triangle)\|_2^2 \geq \kappa(\mathscr{X})\|\triangle\|_F^2,$$

*where $\triangle \in \mathbb{R}^{I_1 \times \cdots \times I_N}$ is an arbitrary tensor.*

**Theorem 5.** *Assume the operator $\mathscr{X}$ satisfies the RSC condition with a constant $\kappa(\mathscr{X})$ and the observations $\mathbf{y} = \mathscr{X}(\mathcal{D}) + \varepsilon$. Let $(\mathcal{G}, U_1, U_2, \ldots, U_N)$ be a KKT point of the following problem with given ranks $R_1 = \cdots = R_N = R$,*

$$\min_{\mathcal{G}, \{U_n \in St(I_n, R_n)\}} \sum_{n=1}^{N} \|\mathcal{G}_{(n)}\|_* + \frac{\lambda}{2}\|\mathbf{y} - \mathscr{X}(\mathcal{G}_{\times 1}U_{1\times} \cdots_{\times N} U_N)\|_2^2. \tag{22}$$

*Then*

$$\text{RMSE} \leq \frac{\|\varepsilon\|_2}{\sqrt{m\kappa(\mathscr{X})I^N}} + \frac{N\sqrt{R}}{C_2\lambda\sqrt{m\kappa(\mathscr{X})I^N}}, \tag{23}$$

*where $C_2 = \frac{\|\mathcal{P}_S\mathscr{X}^*(\mathbf{y}-\mathscr{X}(\mathcal{A}))\|_F}{\|\mathbf{y}-\mathscr{X}(\mathcal{A})\|_2}$ and $\mathscr{X}^*$ denotes the adjoint operator of $\mathscr{X}$.*

The proof of the theorem can be found in Appendix E of the supplementary material.

## 5 Experiments

### 5.1 Synthetic Tensor Completion

Following [17], we generated a low-$n$-rank tensor $\mathcal{T} \in \mathbb{R}^{I_1 \times I_2 \times \cdots \times I_N}$ which we used as the ground truth data. The order of the tensors varies from three to five, and $r$ is set to 10. Furthermore, we randomly sample a few entries from $\mathcal{T}$ and recover the whole tensor with various sampling ratios (SRs) by our gHOI method and the state-of-the-art LRTC algorithms including WTucker [8], WCP [1], FaLRTC [17], and Latent [24]. The relative square error (RSE) of the recovered tensor $\mathcal{X}$ for all these algorithms is defined as RSE := $\|\mathcal{X} - \mathcal{T}\|_F/\|\mathcal{T}\|_F$.

The average results (RSE and running time) of 10 independent runs are shown in Table 1, where the order of tensor data varies from four to five. It is clear that our gHOI method consistently yields much more accurate solutions, and outperforms the other algorithms in terms of both RSE and efficiency. Moreover, we present the running time of our gHOI method and the other methods with varying sizes of third-order tensors, as shown in Fig. 1(a). We can see that the running time of WTcuker, WCP, Latent and FaLRTC dramatically grows with the increase of tensor size whereas that of our gHOI method only increases slightly. This shows that our gHOI method has very good scalability and can address large-scale problems. To further evaluate the robustness of our gHOI method with respect to the given tensor rank changes, we conduct some experiments on the synthetic data of size $100 \times 100 \times 100$, and illustrate the recovery results of all methods with 20% SR, where the rank parameter of gHOI, WTucker and WCP is chosen from $\{10, 15, \cdots, 40\}$. The average RSE results of 10 independent runs are shown in Fig. 1(b), from which we can see that our gHOI method performs much more robust than both WTucker and WCP.

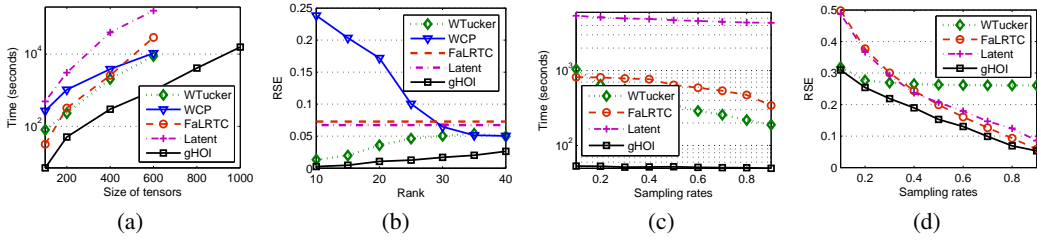

Figure 1: Comparison of all these methods in terms of computational time (in seconds and in logarithmic scale) and RSE on synthetic third-order tensors by varying tensor sizes (a) or given ranks (b), and the BRAINIX data set: running time (c) and RSE (d).

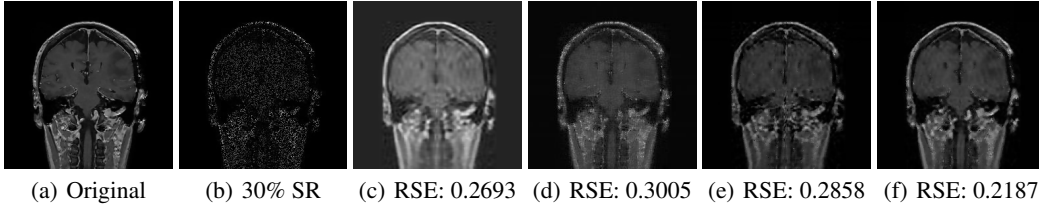

(a) Original   (b) 30% SR   (c) RSE: 0.2693   (d) RSE: 0.3005   (e) RSE: 0.2858   (f) RSE: 0.2187

Figure 2: The recovery results on the BRAINIX data set with 30% SR: (c)-(e) The results of WTucker, FaLRTC, Latent and gHOI, respectively (Best viewed zoomed in).

## 5.2   Medical Images Inpainting

In this part, we apply our gHOI method for medical image inpainting problems on the BRAINIX data set[1]. The recovery results on one randomly chosen image with 30% SR are illustrated in Fig. 2. Moreover, we also present the recovery accuracy (RSE) and running time (seconds) with varying SRs, as shown in Fig. 1(c) and (d). From these results, we can observe that our gHOI method consistently performs better than the other methods in terms of both RSE and efficiency. Especially, gHOI is about 20 times faster than WTucker and FaLRTC, and more than 90 times faster than Latent, when the sample percentage is 10%. By increasing the sampling rate, the RSE results of three Schatten 1-norm minimization methods including Latent, FaLRTC and gHOI, dramatically reduce. In contrast, the RSE of WTucker decreases slightly.

## 6   Conclusions

We proposed a scalable core tensor Schatten 1-norm minimization method for simultaneous tensor decomposition and completion. First, we induced the equivalence relation of the Schatten 1-norm of a low-rank tensor and its core tensor. Then we formulated a tractable Schatten 1-norm regularized tensor decomposition model with missing data, which is a convex combination of multiple much smaller-scale matrix SNM. Finally, we developed an efficient gHOI algorithm to solve our problem. Moreover, we also provided the convergence analysis and recovery guarantee of our algorithm. The convincing experimental results verified the efficiency and effectiveness of our gHOI algorithm. gHOI is significantly faster than the state-of-the-art LRTC methods. In the future, we will apply our gHOI algorithm to address a variety of robust tensor recovery and completion problems, e.g., higher-order RPCA [10] and robust LRTC.

## Acknowledgments

This research is supported in part by SHIAE Grant No. 8115048, MSRA Grant No. 6903555, GRF No. 411211, CUHK direct grant Nos. 4055015 and 4055017, China 973 Fundamental R&D Program, No. 2014CB340304, and Huawei Grant No. 7010255.

## Footnotes

[1]http://www.osirix-viewer.com/datasets/

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
