[Supplementary Material]

# Supplementary Materials for "Generalized Higher-Order Orthogonal Iteration for Tensor Decomposition and Completion"

**Yuanyuan Liu[†], Fanhua Shang[‡][*], Wei Fan[§], James Cheng[‡], Hong Cheng[†]**

[†]Dept. of Systems Engineering and Engineering Management,
The Chinese University of Hong Kong

[‡]Dept. of Computer Science and Engineering, The Chinese University of Hong Kong

[§]Huawei Noah′s Ark Lab, Hong Kong

{yyliu, hcheng}@se.cuhk.edu.hk {fhshang, jcheng}@cse.cuhk.edu.hk
david.fanwei@huawei.com

In this supplementary material, we give some additional experimental results on the Ocean video, graph regularized extensions, and detailed proofs of some lemmas and theorems.

## Notations

**Definition 4.** *The Kronecker product $A \otimes B \in \mathbb{R}^{mp \times nq}$ of two matrices $A \in \mathbb{R}^{m \times n}$ and $B \in \mathbb{R}^{p \times q}$ is defined as*

$$A \otimes B = \begin{bmatrix} a_{1,1}B & a_{1,2}B & \cdots & a_{1,n}B \\ a_{2,1}B & a_{2,2}B & \cdots & a_{2,n}B \\ \vdots & \vdots & \ddots & \vdots \\ a_{m,1}B & a_{m,2}B & \cdots & a_{m,n}B \end{bmatrix}.$$

**Definition 5.** *The Hadamard product $A \odot B \in \mathbb{R}^{m \times n}$ of two same-sized matrices $A, B \in \mathbb{R}^{m \times n}$ is given by*

$$A \odot B = \begin{bmatrix} a_{1,1}b_{1,1} & a_{1,2}b_{1,2} & \cdots & a_{1,n}b_{1,n} \\ a_{2,1}b_{2,1} & a_{2,2}b_{2,2} & \cdots & a_{2,n}b_{2,n} \\ \vdots & \vdots & \ddots & \vdots \\ a_{m,1}b_{m,1} & a_{m,2}b_{m,2} & \cdots & a_{m,n}b_{m,n} \end{bmatrix}.$$

*The definition can be extended to N-th order tensors by the following form,*

$$(\mathcal{A} \odot \mathcal{B})_{i_1,i_2,\cdots,i_N} = a_{i_1,i_2,\cdots,i_N} b_{i_1,i_2,\cdots,i_N}.$$

## 1 More Experimental Results

We test our gHOI method for color video inpainting problems, and compare gHOI against WTucker[1], WCP[2], FaLRTC[3] and Latent[4] on the Ocean video used in [2]. The color videos are naturally represented as a fourth-order tensor (i.e., length×width×channels×frames), whose size is $112 \times 160 \times 3 \times 32$. For WTucker and our gHOI method, the tensor ranks are set to

---

[*]Corresponding author.

[1]http://www.lair.irb.hr/ikopriva/marko-filipovi.html

[2]http://www.sandia.gov/~tgkolda/TensorToolbox/

[3]http://pages.cs.wisc.edu/~ji-liu/

[4]http://ttic.uchicago.edu/~ryotat/softwares/tensor/

Figure 1: Recovery results on the Ocean video with 20% SR. From left to right in the first row: input color image and the reconstruction results of WTucker and WCP. From left to right in the second row: the results of FaLRTC, Latent and gHOI, respectively (Best viewed in color).

Table 1: RSE and time cost (seconds) comparison on the Ocean video.

|  | WTucker | WCP | FaLRTC | Latent | gHOI |
|---|---|---|---|---|---|
| RSE±std | 0.1109±0.0005 | 0.1852±0.0084 | 0.1174±0.0006 | 0.1438±0.0011 | **0.1041±0.0002** |
| Time±std | 381.46±11.26 | 230.48±6.19 | 130.82±5.53 | 1490.76±20.81 | **31.72±1.43** |

$R_1 = R_2 = R_4 = 20$ and $R_3 = 3$, and the regularization parameter $\lambda = 100$. For FaLRTC and our gHOI method, the weights $\alpha_n$ ($n = 1, \ldots, 4$) are set to be $\{1, 1, 10^{-3}, 10^{-3}\}$. In addition, the smoothing parameters of FaLRTC are set to be $5\alpha_n/I_n$ for $n = 1, \ldots, 4$. The tolerance value of all these methods is fixed at tol $= 10^{-4}$. The recovery results on one randomly chosen frame with 20% SR are shown in Fig. 1. Moreover, we also report the recovery accuracy (RSE) and running time (seconds) in Table 1. From these results, we can observe that our gHOI method consistently performs better than the other methods in terms of accuracy and efficiency.

## 2  Graph Regularization Extensions

### 2.1  Graph Regularized Model

As our gHOI method is a tensor decomposition method, and inspired by the work [3], we exploit the auxiliary information given as similarity matrices in a regularization model:

$$\min_{\mathcal{G},\{U_n\},\mathcal{X}} \sum_{n=1}^{N} \|\mathcal{G}_{(n)}\|_* + \frac{\lambda}{2}\|\mathcal{X} - \mathcal{G} \times_1 U_1 \cdots \times_N U_N\|_F^2 + \gamma \text{Tr}(\sum_{n=1}^{N} U_n^T L_n U_n),$$

$$\text{s.t.,} \ \mathcal{P}_\Omega(\mathcal{X}) = \mathcal{P}_\Omega(\mathcal{T}), \ U_n \in \text{St}(I_n, R_n), \ n = 1, \cdots, N,$$

where $\gamma \geq 0$ is a regularization constant, $\text{Tr}(\cdot)$ denotes the matrix trace, $L_n$ is the graph Laplacian matrix, i.e., $L_n = D_n - W_n$, $W_n$ is the weight matrix for the object set $S_n$, and $D_n$ is the diagonal matrix whose entries are column sums of $W_n$, i.e., $(D_n)_{ii} = \sum_j (W_n)_{ij}$. Moreover, Algorithm 1 can be extended to solve our graph regularized tensor completion problem.

### 2.2  Multi-Relational Learning Applications

In this part, we test our graph regularized gHOI (RgHOI) method for link prediction problems on a real-world network data set, YouTube[5] [4]. YouTube is currently the most popular video sharing web site, which allows users to interact with each other in various forms such as contacts, subscriptions, sharing favorite videos, etc. In total, this data set contains 848,003 users, with 15,088 users sharing all of the information types, and includes 5-dimensions of interactions: contact network,

Figure 2: Running time (in seconds and in logarithmic scale) comparison of RWTucker, RWCP, FaLRTC and RgHOI on the YouTube data set. For each dataset, we use 20% for training. Note that RWTucker, RWCP and FaLRTC could not run for sizes $\{8,000, 15,088\}$ due to runtime exceptions.

Figure 3: ROC curves showing the performance of link prediction methods with 10% (left) and 20% (right) training data, respectively (Best viewed in color).

co-contact network, co-subscription network, co-subscribed network, and favorite network. We run these experiments on a machine with 6-core Intel Xeon 2.4GHz CPU and 64GB memory.

For the graph regularized WTucker (RWTucker) and WCP (RWCP), and our RgHOI method, we set the tensor ranks $R_1 = R_2 = 40$ and $R_3 = 5$, and $\lambda = 100$. For FaLRTC and RgHOI, $\alpha_n$ $(n = 1, 2, 3)$ are set to be $\{1, 1, 10^{-3}\}$. The smoothing parameters of FaLRTC are set to be $5\alpha_n/I_n$ for $n = 1, 2, 3$. The tolerance value of all these methods is fixed at tol $= 10^{-4}$.

We use the 15,088 users who share all of the information types and have 5-dimensions of interactions in our experiments. So the size of the input tensor is $15,088 \times 15,088 \times 5$. We first report the average running time (in seconds) of three graph regularized algorithms including RWTucker [3], RWCP [3] and our RgHOI method, and FaLRTC over 10 independent runs in Fig. 2, where the number of users is gradually increased. Our RgHOI method is much faster than RWTucker, RWCP and FaLRTC. The running time of our RgHOI method increases only slightly when the number of users increases. On the contrary, the running time of RWTucker, RWCP and FaLRTC increases dramatically. They could not yield experimental results within 48 hours when the number of users is 8,000 or 15,088. This shows that our RgHOI method has very good scalability and can address large-scale problems.

As the other methods cannot finish running when the problem size is large, we choose 4,117 users who have more than 10 interactions to form a subset of size $4,117 \times 4,117 \times 5$. We randomly select 10% or 20% entries as the training set, and the remainder as the testing data. We illustrate the average prediction accuracy (the score Area Under the receiver operating characteristic Curve, AUC) over 10 independent runs in Fig. 3. From the results, we can see that our RgHOI method significantly outperforms the state-of-the-art LRTC methods in terms of both effectiveness and efficiency.

# Appendix A: Proof of Theorem 1

Before giving the proof of Theorem 1, we will first present some properties of matrices and tensors in the following.

**Property 1.** *Let $X \in \mathbb{R}^{m \times l}$, $Y \in \mathbb{R}^{n \times l}$, and $Z \in \mathbb{R}^{l \times l}$, then*

$$\|XZY^T\|_* = \|Z\|_*,$$

*where both $X$ and $Y$ are column-orthonormal, i.e., $X \in St(m, l)$ and $Y \in St(n, l)$.*

**Property 2.** *Let $A \in \mathbb{R}^{m \times n}$, $B \in \mathbb{R}^{p \times q}$, and $C$ and $D$ be two matrices of suitable sizes, then we have the following results:*

**I.** $(A \otimes B) \otimes C = A \otimes (B \otimes C)$.

**II.** $(A \otimes B)(C \otimes D) = (AC) \otimes (BD)$.

**III.** $(A \otimes B)^T = A^T \otimes B^T$.

**Property 3.** *Let $\mathcal{X} = \mathcal{G} \times_1 U_1 \times_2 \cdots \times_N U_N$, where $\mathcal{X} \in \mathbb{R}^{I_1 \times I_2 \cdots \times I_N}$, $\mathcal{G} \in \mathbb{R}^{R_1 \times R_2 \cdots \times R_N}$, and $U_n \in \mathbb{R}^{I_n \times R_n}$, $n = 1, \ldots, N$, then, for any $n \in \{1, \ldots, N\}$, we have*

$$\mathcal{X}_{(n)} = U_n \mathcal{G}_{(n)} (U_N \otimes \ldots \otimes U_{n+1} \otimes U_{n-1} \otimes \ldots \otimes U_1)^T.$$

**Proof of Theorem 1:**

*Proof.*
Let $Q_n = U_N \otimes \ldots \otimes U_{n+1} \otimes U_{n-1} \otimes \ldots \otimes U_1$ and $U_n \in St(I_n, R_n)$, $n = 1, \ldots, N$, and according to Property 2, we have the following conclusion:

$Q_n^T Q_n$
$= (U_N \otimes \ldots \otimes U_{n+1} \otimes U_{n-1} \otimes \ldots \otimes U_1)^T (U_N \otimes \ldots \otimes U_{n+1} \otimes U_{n-1} \otimes \ldots \otimes U_1)$
$= (U_N^T \otimes \ldots \otimes U_{n+1}^T \otimes U_{n-1}^T \otimes \ldots \otimes U_1^T)(U_N \otimes \ldots \otimes U_{n+1} \otimes U_{n-1} \otimes \ldots \otimes U_1)$
$= (U_N^T \otimes \ldots \otimes U_{n+1}^T \otimes U_{n-1}^T \otimes \ldots \otimes U_2^T)(U_N \otimes \ldots \otimes U_{n+1} \otimes U_{n-1} \otimes \ldots \otimes U_2) \otimes (U_1^T U_1)$
$= (U_N^T \otimes \ldots \otimes U_{n+1}^T \otimes U_{n-1}^T \otimes \ldots \otimes U_2^T)(U_N \otimes \ldots \otimes U_{n+1} \otimes U_{n-1} \otimes \ldots \otimes U_2) \otimes I_{R_1}$
$= (U_N^T \otimes \ldots \otimes U_{n+1}^T \otimes U_{n-1}^T \otimes \ldots \otimes U_3^T)(U_N \otimes \ldots \otimes U_{n+1} \otimes U_{n-1} \otimes \ldots \otimes U_3) \otimes (U_2^T U_2) \otimes I_{R_1}$
$\vdots$
$= I_{R_N} \otimes \ldots \otimes I_{R_{n+1}} \otimes I_{R_{n-1}} \otimes \ldots \otimes I_{R_2} \otimes I_{R_1}$
$= I_{J_n},$

where $I_{R_i} \in \mathbb{R}^{R_i \times R_i}$ $(i = 1, \ldots, N)$ are all identity matrices, $I_{J_n} \in \mathbb{R}^{J_n \times J_n}$ is also an identity matrix, and $J_n = \Pi_{j \neq n} R_j$.

Then by Property 3, we have

$$\|\mathcal{X}_{(n)}\|_* = \|U_n \mathcal{G}_{(n)}(U_N \otimes \ldots U_{n+1} \otimes U_{n-1} \ldots \otimes U_1)^T\|_*.$$

According to Properties 1 and 3, and $Q_n^T Q_n = I_{J_n}$, we obtain

$$\|\mathcal{X}_{(n)}\|_* = \|U_n \mathcal{G}_{(n)}(U_N \otimes \ldots U_{n+1} \otimes U_{n-1} \ldots \otimes U_1)^T\|_* = \|\mathcal{G}_{(n)}\|_*.$$

Hence, $\|\mathcal{X}\|_* = \|\mathcal{G}\|_*$. $\qquad\square$

## Appendix B: Proof of Theorem 2

*Proof.* Let

$$f(\mathcal{G}, U_1, \ldots, U_N) = \sum_{n=1}^{N} \frac{\mu^k}{2} \|\mathcal{G}_{(n)} - V_n^k + Y_n^k/\mu^k\|_F^2 + \frac{\lambda}{2} \|\mathcal{X}^k - \mathcal{G} \times_1 U_1 \cdots \times_N U_N\|_F^2, \quad (24)$$

then the closed-form solution of (24) with respect to $\mathcal{G}$ is given by

$$\mathcal{G} = \frac{1}{\lambda + N\mu^k} (\lambda \mathcal{M} + \mu^k \mathcal{N}). \quad (25)$$

By (25), and according to the definitions of the tensors $\mathcal{M}$ and $\mathcal{N}$, we obtain

$$\begin{aligned}
&\langle \mathcal{X}^k, \ \mathcal{G} \times_1 U_1 \cdots \times_N U_N \rangle \\
=&\langle \mathcal{X}^k \times_1 (U_1)^T \cdots \times_N (U_N)^T, \ \mathcal{G} \rangle \\
=&\left\langle \mathcal{M}, \ \frac{1}{\lambda + N\mu^k} (\lambda \mathcal{M} + \mu^k \mathcal{N}) \right\rangle \\
=&\frac{\lambda}{\lambda + N\mu^k} \|\mathcal{M}\|_F^2 + \frac{\mu^k}{\lambda + N\mu^k} \langle \mathcal{M}, \ \mathcal{N} \rangle,
\end{aligned} \quad (26)$$

and

$$\begin{aligned}
&\langle \mathcal{G}, \ \mathcal{N} \rangle \\
=&\left\langle \frac{1}{\lambda + N\mu^k} (\lambda \mathcal{M} + \mu^k \mathcal{N}), \ \mathcal{N} \right\rangle \\
=&\frac{\lambda}{\lambda + N\mu^k} \langle \mathcal{M}, \ \mathcal{N} \rangle + \frac{\mu^k}{\lambda + N\mu^k} \|\mathcal{N}\|_F^2.
\end{aligned} \quad (27)$$

Hence, $f(\mathcal{G}, U_1, U_2, \ldots, U_N)$ is rewritten as follows:

$$\begin{aligned}
&f(\mathcal{G}, U_1, \ldots, U_N) \\
=&\frac{\lambda}{2} \|\mathcal{X}^k\|_F^2 - \lambda \langle \mathcal{X}^k, \mathcal{G} \times_1 U_1 \cdots \times_N U_N \rangle + \frac{\lambda}{2} \|\mathcal{G}\|_F^2 \\
&+ \frac{N\mu^k}{2} \|\mathcal{G}\|_F^2 - \mu^k \langle \mathcal{G}, \ \mathcal{N} \rangle + \frac{\mu^k}{2} \sum_{n=1}^{N} \|V_n^k - Y_n^k/\mu^k\|_F^2.
\end{aligned} \quad (28)$$

Substituting (25), (26) and (27) into (28), then the cost function (28) is formulated in the following form,

$$\begin{aligned}
&f(\mathcal{G}, U_1, \ldots, U_N) \\
=&\frac{\lambda}{2} \|\mathcal{X}^k\|_F^2 - \lambda \langle \mathcal{M}, \ \mathcal{G} \rangle + \frac{\lambda + N\mu^k}{2} \|\mathcal{G}\|_F^2 - \mu^k \langle \mathcal{G}, \ \mathcal{N} \rangle \\
&+ \frac{\mu^k}{2} \sum_{n=1}^{N} \|V_n^k - Y_n^k/\mu^k\|_F^2 \\
=&\zeta - \frac{1}{2(\lambda + N\mu^k)} \|\lambda \mathcal{M} + \mu^k \mathcal{N}\|_F^2 \\
=&\zeta - \frac{1}{2(\lambda + N\mu^k)} g(U_1, U_2, \ldots, U_N),
\end{aligned}$$

where $\zeta = \frac{\lambda}{2} \|\mathcal{X}^k\|_F^2 + \frac{\mu^k}{2} \sum_{n=1}^{N} \|V_n^k - Y_n^k/\mu^k\|_F^2$ is a constant with respect to $\{\mathcal{G}, U_1, \ldots, U_N\}$. Combination with the above results proves the theorem. $\square$

## Appendix C: Proof of Theorem 3

The proof sketch of Theorem 3 is similar to that in [1]. We first prove the boundedness of multipliers and some variables of Algorithm 1, and then we analyze the convergence of Algorithm 1. To prove the boundedness, we first give the following lemmas.

**Lemma 1** ([1]). *Let $\mathcal{X}$ be a real Hilbert space endowed with an inner product $\langle \cdot \rangle$ and a corresponding norm $\| \cdot \|$ (e.g., the Schatten 1-norm), and $y \in \partial \|x\|$, where $\partial \| \cdot \|$ denotes the subgradient of the norm. Then $\|y\|^* = 1$ if $x \neq 0$, and $\|y\|^* \leq 1$ if $x = 0$, where $\| \cdot \|^*$ is the dual norm of $\| \cdot \|$.*

**Lemma 2.** *Let $Y_n^{k+1} = Y_n^k + \mu^k(\mathcal{G}_{(n)}^{k+1} - V_n^{k+1})$ for any $n \in \{1, \ldots, N\}$, then the sequences $\{\mathcal{G}^{k+1}\}$, $\{Y_n^{k+1}\}$ and $\{V_n^{k+1}\}$ $(n = 1, \ldots, N)$ produced by Algorithm 1 are bounded.*

*Proof.* Let $\mathcal{U}^{k+1} = (U_1^{k+1}, U_2^{k+1}, \ldots, U_N^{k+1})$, $\mathcal{V}^{k+1} = (V_1^{k+1}, V_2^{k+1}, \ldots, V_N^{k+1})$ and $\mathcal{Y}^{k+1} = (Y_1^{k+1}, Y_2^{k+1}, \ldots, Y_N^{k+1})$. By the optimality condition of (10), we have

$$0 \in \partial_{V_n} \mathcal{L}_{\mu^k}(\mathcal{X}^k, \mathcal{G}^{k+1}, \mathcal{U}^{k+1}, \mathcal{V}^{k+1}, \mathcal{Y}^k), \ \forall n \in \{1, \cdots, N\},$$

i.e.,

$$Y_n^k + \mu^k(\mathcal{G}_{(n)}^{k+1} - V_n^{k+1}) \in \partial \|V_n^{k+1}\|_*, \ \forall n \in \{1, \cdots, N\},$$

and

$$Y_n^{k+1} \in \partial \|V_n^{k+1}\|_* \ \forall n \in \{1, \cdots, N\}.$$

By Lemma 1, we have

$$\|Y_n^{k+1}\|_2 \leq 1, \ \forall n \in \{1, \cdots, N\},$$

where $\| \cdot \|_2$ is the spectral norm, which is equal to the largest singular value of the matrix. Hence, $\{Y_n^k\}$, $n = 1, \ldots, N$, are bounded.

By (13), we obtain

$$\|\mathcal{X}^{k+1} - \mathcal{G}^{k+1} \times_1 U_1^{k+1} \cdots \times_N U_N^{k+1}\|_F^2$$
$$= \|\mathcal{P}_\Omega(\mathcal{T} - \mathcal{G}^{k+1} \times_1 U_1^{k+1} \cdots \times_N U_N^{k+1})\|_F^2$$
$$\leq \|\mathcal{P}_\Omega(\mathcal{T} - \mathcal{G}^{k+1} \times_1 U_1^{k+1} \cdots \times_N U_N^{k+1})\|_F^2 + \|\mathcal{P}_{\Omega^C}(\mathcal{X}^k - \mathcal{G}^{k+1} \times_1 U_1^{k+1} \cdots \times_N U_N^{k+1})\|_F^2$$
$$= \|\mathcal{X}^k - \mathcal{G}^{k+1} \times_1 U_1^{k+1} \cdots \times_N U_N^{k+1}\|_F^2.$$

By the iteration procedure, we have

$$\mathcal{L}_{\mu^k}(\mathcal{X}^{k+1}, \mathcal{G}^{k+1}, \mathcal{U}^{k+1}, \mathcal{V}^{k+1}, \mathcal{Y}^k)$$
$$\leq \mathcal{L}_{\mu^k}(\mathcal{X}^k, \mathcal{G}^{k+1}, \mathcal{U}^{k+1}, \mathcal{V}^k, \mathcal{Y}^k)$$
$$\leq \mathcal{L}_{\mu^k}(\mathcal{X}^k, \mathcal{G}^k, \mathcal{U}^k, \mathcal{V}^k, \mathcal{Y}^k)$$
$$= \mathcal{L}_{\mu^{k-1}}(\mathcal{X}^k, \mathcal{G}^k, \mathcal{U}^k, \mathcal{V}^k, \mathcal{Y}^{k-1}) + \beta_k \Sigma_{n=1}^N \|Y_n^k - Y_n^{k-1}\|_F^2,$$

where $\beta_k = \frac{\mu^{k-1} + \mu^k}{2(\mu^{k-1})^2}$ and $\mu^k = \rho \mu^{k-1}$.

Since

$$\sum_{k=1}^{\infty} \frac{\mu^{k-1} + \mu^k}{2(\mu^{k-1})^2} = \frac{\rho(\rho+1)}{2\mu^0(\rho-1)} < \infty,$$

$\{\mathcal{L}_{\mu^{k-1}}(\mathcal{X}^k, \mathcal{G}^k, \mathcal{U}^k, \mathcal{V}^k, \mathcal{Y}^{k-1})\}$ is bounded due to the boundedness of $\{Y_n^k\}$ for $n = 1, \ldots, N$.

$$\sum_{n=1}^N \|V_n^k\|_* + \frac{\lambda}{2} \|\mathcal{X}^k - \mathcal{G}^k \times_1 U_1^k \times_2 \cdots \times_N U_N^k\|_F^2$$

$$= \mathcal{L}_{\mu^{k-1}}(\mathcal{X}^k, \mathcal{G}^k, \mathcal{U}^k, \mathcal{V}^k, \mathcal{Y}^{k-1}) - \frac{1}{2} \sum_{n=1}^N \frac{\|Y_n^k\|_F^2 - \|Y_n^{k-1}\|_F^2}{\mu^{k-1}}$$

is upper bounded, hence $\{\mathcal{G}^k\}$, $\{\mathcal{X}^k\}$ and $\{V_n^k\}$, $n = 1, \ldots, N$, are bounded. $\qquad \square$

**Proof of Theorem 3:**

*Proof.* **I.** By $(\mathcal{G}_{(n)}^{k+1} - V_n^{k+1}) = (\mu^k)^{-1}(Y_n^{k+1} - Y_n^k)$, the boundedness of $\{Y_n^k\}$ and $\lim_{k\to\infty} \mu^k = \infty$, we have

$$\lim_{k\to\infty} \|\mathcal{G}_{(n)}^{k+1} - V_n^{k+1}\|_F = 0, \ \ \forall n \in \{1,\cdots,N\}.$$

Thus, $(\mathcal{G}^k, V_n^k, \mathcal{X}^k)$ approaches to a feasible solution.

Next we prove that the sequences $\{\mathcal{G}^k\}$ and $\{V_n^k\}$, $n = 1, \ldots, N$, are Cauchy sequences.

By (20), we have

$$\mathcal{G}^{k+1} - \mathcal{G}^k$$
$$= \frac{1}{\lambda + N\mu^k}(\lambda \mathcal{X}^k \times_1 (U_1^{k+1})^T \cdots \times_N (U_N^{k+1})^T)$$
$$+ \frac{\mu^k}{\lambda + N\mu^k} \sum_{n=1}^{N} \mathrm{refold}(V_n^k - Y_n^k/\mu^k - \mathcal{G}_{(n)}^k) - \frac{\lambda}{\lambda + N\mu^k}\mathcal{G}^k.$$

By the result above, we have $\|\mathcal{G}_{(n)}^k - V_n^k\|_F = O((\mu^k)^{-1})$. Since $\{\mathcal{G}^k\}$ is bounded, then we have $\frac{\lambda}{\lambda+N\mu^k}\mathcal{G}^k \to 0$. $\|\mathcal{G}^{k+1} - \mathcal{G}^k\|_F = O((\mu^k)^{-1})$, and then $\sum_{k=1}^{\infty}(\mu^{k-1})^{-1} = \frac{\rho}{\mu^0(\rho-1)} < \infty$. Hence, $\{\mathcal{G}^k\}$ is a Cauchy sequence, and it has a limit, $\mathcal{G}^{\infty}$.

Similarly, $\{V_n^k\}$ $(n = 1, \ldots, N)$ are also Cauchy sequences, therefore they have their limits, $V_n^{\infty}$, respectively.

**II.** The Karush-Kuhn-Tucker (KKT) conditions of (6) are formulated as follows:

$$0 \in \sum_{n=1}^{N} \mathrm{refold}(\partial\|\mathcal{G}_{(n)}^*\|_*) + \lambda(\mathcal{G}^* - \mathcal{X}^* \times_1 (U_1^*)^T \cdots \times_N (U_N^*)^T),$$
$$\mathcal{X}_\Omega^* = T_\Omega, \quad \mathcal{X}_{\Omega^C}^* = (\mathcal{G}^* \times_1 U_1^* \cdots \times_N U_N^*)_{\Omega^C},$$
$$U_n^* \in \mathrm{St}(I_n, R_n), \ n = 1, \ldots, N.$$

According to Algorithm 1, the first-order optimal condition of the problem (10) at the $k$-th iteration is formulated as follows:

$$0 \in \sum_{n=1}^{N} \mathrm{refold}(\partial\|V_n^{k+1}\|_*) + N\mu^k\mathcal{G}^{k+1} - \mu^k\sum_{n=1}^{N}\mathrm{refold}(V_n^{k+1} - Y_n^k/\mu^k). \tag{29}$$

The first-order optimal condition of the problems (9) with respect to $\mathcal{G}$ is

$$0 = \sum_{n=1}^{N}(\lambda+N\mu^k)\mathcal{G}^{k+1} - \mu^k\sum_{n=1}^{N}\mathrm{refold}(V_n^k - Y_n^k/\mu^k) - \lambda(\mathcal{X}^k \times_1 (U_1^{k+1})^T \cdots \times_N (U_N^{k+1})^T). \tag{30}$$

Since $\{V_n^k\}$, $n = 1, \ldots, N$, and $\{\mathcal{G}^k\}$ are Cauchy sequences, and let $V_n^{\infty}$ $(n = 1, \ldots, N)$ and $\mathcal{G}^{\infty}$ be their limit points, respectively. By the result (**I**), we have $V_n^{\infty} = \mathcal{G}_n^{\infty}$ for all $n = 1, \ldots, N$. By (29) and (30), we obtain

$$0 \in \sum_{n=1}^{N}\mathrm{refold}(\partial\|\mathcal{G}_{(n)}^{\infty}\|_*) + \lambda(\mathcal{G}^{\infty} - \mathcal{X}^{\infty} \times_1 (U_1^{\infty})^T \cdots \times_N (U_N^{\infty})^T).$$

Furthermore, by (18) and (13), we have

$$\mathcal{X}_\Omega^{\infty} = T_\Omega, \quad \mathcal{X}_{\Omega^C}^{\infty} = (\mathcal{G}^{\infty} \times_1 U_1^{\infty} \cdots \times_N U_N^{\infty})_{\Omega^C},$$
$$U_n^{\infty} \in \mathrm{St}(I_n, R_n), \ n = 1, \ldots, N. \tag{31}$$

Hence, the sequence $\{\mathcal{G}^k, \mathcal{X}^k, U_1^k, \ldots, U_N^k\}$ generated by Algorithm 1 converges the KKT point of (6). $\square$

## Appendix D: Proof of Theorem 4

We extend the results in [5] to our tensor completion and decomposition model (6) with Schatten 1-norm regularization. By substituting (31) into (6), then the minimization problem (6) is trivially equivalent to

$$\min_{\mathcal{G},\{U_n\}} \sum_{n=1}^{N} \|\mathcal{G}_{(n)}\|_* + \frac{\lambda}{2}\|\mathcal{P}_\Omega(\mathcal{T}) - \mathcal{P}_\Omega(\mathcal{G} \times_1 U_1 \cdots \times_N U_N)\|_F^2. \tag{32}$$

According to Theorem 3, we can know that $(\mathcal{G}, U_1, \cdots, U_N)$ is also a KKT point of the problem (32). To prove Theorem 4, we first give the following lemma [5].

**Lemma 3.** *Let* $\mathcal{L}(X) = \frac{1}{\sqrt{mn}}\|X - \widehat{X}\|_F$ *and* $\hat{\mathcal{L}}(X) = \frac{1}{\sqrt{|\Omega|}}\|\mathcal{P}_\Omega(X - \widehat{X})\|_F$ *be the actual and empirical loss function respectively, where* $X, \widehat{X} \in \mathbb{R}^{m \times n}$ $(m \leq n)$. *Furthermore, assume entrywise constraint* $\max_{i,j} |X_{ij}| \leq \beta_1$. *Then for all rank-r matrices* $X$, *with probability greater than* $1 - 2\exp(-n)$, *there exists a fixed constant* $C_3$ *such that*

$$\sup_{X \in S_r} |\hat{\mathcal{L}}(X) - \mathcal{L}(X)| \leq C_3 \beta_1 \left(\frac{nr \log(n)}{|\Omega|}\right)^{1/4},$$

*where* $S_r = \{X \in \mathbb{R}^{m \times n} : rank(X) \leq r, \|X\|_F \leq \sqrt{mn}\beta_1\}$.

Here we set $M = \max_{i,j}(X_{ij} - \widehat{X}_{ij})^2 \leq (2\beta_1)^2$ and $\epsilon = 9\beta_1$ as in [5]. According to Theorem 2 in [5], thus we have

$$\sup_{X \in S_r} |\hat{\mathcal{L}}(X) - \mathcal{L}(X)|$$

$$\leq \frac{2\epsilon}{\sqrt{|\Omega|}} + \left(\frac{M^2}{2}\frac{2nr \log(9\beta_1 n/\epsilon)}{|\Omega|}\right)^{1/4}$$

$$\leq \frac{18\beta_1}{\sqrt{|\Omega|}} + 2\beta_1 \left(\frac{nr \log(n)}{|\Omega|}\right)^{1/4}$$

$$= \left(2 + \frac{18}{(|\Omega|nr \log(n))^{1/4}}\right) \beta_1 \left(\frac{nr \log(n)}{|\Omega|}\right)^{1/4}.$$

Hence, $C_3$ can be set to $2 + \frac{18}{(|\Omega|nr \log(n))^{1/4}}$.

**Proof of Theorem 4:**

*Proof.* Let $\mathcal{A} = \mathcal{G}_{\times 1}U_{1\times}\ldots_{\times N}U_N$, we first need to bound $\|\mathcal{T} - \mathcal{A}\|_F$. By $C_1 = \frac{\|\mathcal{P}_S\mathcal{P}_\Omega(\mathcal{T}-\mathcal{A})\|_F}{\|\mathcal{P}_\Omega(\mathcal{T}-\mathcal{A})\|_F}$, we have

$$\frac{\|\mathcal{T} - \mathcal{A}\|_F}{\sqrt{I^N}}$$

$$\leq \left|\frac{\|\mathcal{T} - \mathcal{A}\|_F}{\sqrt{I^N}} - \frac{\|\mathcal{P}_S\mathcal{P}_\Omega(\mathcal{T} - \mathcal{A})\|_F}{C_1\sqrt{|\Omega|}}\right| + \frac{\|\mathcal{P}_S\mathcal{P}_\Omega(\mathcal{T} - \mathcal{A})\|_F}{C_1\sqrt{|\Omega|}}$$

$$= \left|\frac{\|\mathcal{T} - \mathcal{A}\|_F}{\sqrt{I^N}} - \frac{\|\mathcal{P}_\Omega(\mathcal{T} - \mathcal{A})\|_F}{\sqrt{|\Omega|}}\right| + \frac{\|\mathcal{P}_S\mathcal{P}_\Omega(\mathcal{T} - \mathcal{A})\|_F}{C_1\sqrt{|\Omega|}}.$$

Let $\varphi(\Omega) = \left|\frac{1}{\sqrt{I^N}}\|\mathcal{T} - \mathcal{A}\|_F - \frac{1}{\sqrt{|\Omega|}}\|\mathcal{P}_\Omega(\mathcal{T} - \mathcal{A})\|_F\right|$, then we need to bound $\varphi(\Omega)$. According to Lemma 3, we unfold the tensors $\mathcal{T}$ and $\mathcal{A}$ along the $n$-th mode, $\forall n \in \{1, \cdots, N\}$. Since $rank(\mathcal{A}_{(n)}) \leq R$ and $\mathcal{A}_{(n)} \in S_R$, then with probability greater than $1 - 2\exp(-I^{N-1})$, there exists

a fixed constant $C = 2 + \frac{18}{(|\Omega|I^{N-1}R\log(I^{N-1}))^{1/4}}$ such that

$$\sup_{\mathcal{A}_{(n)} \in S_R} \varphi(\Omega) = \left| \frac{\|\mathcal{A}_{(n)} - \mathcal{T}_{(n)}\|_F}{\sqrt{I^N}} - \frac{\|(\mathcal{P}_\Omega(\mathcal{A}))_{(n)} - (\mathcal{P}_\Omega(\mathcal{T}))_{(n)}\|_F}{\sqrt{|\Omega|}} \right|$$

$$\leq C\beta \left( \frac{I^{N-1}R\log(I^{N-1})}{|\Omega|} \right)^{\frac{1}{4}}. \tag{33}$$

Next we need to bound $\|\mathcal{P}_S\mathcal{P}_\Omega(\mathcal{T} - \mathcal{A})\|_F$. Since $(\mathcal{G}, U_1, \cdots, U_N)$ is a KKT point of the problem (32), the first-order optimal condition of the problem (32) with respect to $\mathcal{G}$ is written as follows:

$$\lambda \mathcal{P}_\Omega(\mathcal{T} - \mathcal{A}) \times_1 U_1^T \times \cdots \times_N U_N^T \in \sum_{n=1}^N \text{refold}(\partial\|\mathcal{G}_{(n)}\|_*). \tag{34}$$

In other words, there exist $\{P_n \in \mathbb{R}^{R \times R^{N-1}}, \ n = 1, \cdots, N\}$ such that

$$\lambda P_n \in \partial\|\mathcal{G}_{(n)}\|_*, \ n = 1, 2, \cdots, N, \tag{35a}$$

$$\mathcal{P}_\Omega(\mathcal{T} - \mathcal{A}) \times_1 U_1^T \times \cdots \times_N U_N^T = \sum_{n=1}^N \text{refold}(P_n). \tag{35b}$$

Using Lemma 1 in Appendix C and (35a), we obtain

$$\lambda\|P_n\|_2 \leq 1,$$

where $\|\cdot\|_2$ is the spectral norm. According to $\text{rank}(P_n) \leq R$, we have

$$\|P_n\|_F \leq \sqrt{R}\|P_n\|_2 \leq \frac{\sqrt{R}}{\lambda}. \tag{36}$$

By (35b) and (36), we obtain

$$\|\mathcal{P}_\Omega(\mathcal{T} - \mathcal{A}) \times_1 U_1^T \times \cdots \times_N U_N^T\|_F$$
$$\leq \sum_{n=1}^N \|\text{refold}(P_n)\|_F = \sum_{n=1}^N \|P_n\|_F \tag{37}$$
$$\leq \frac{N\sqrt{R}}{\lambda}.$$

By the definition of $\mathcal{P}_S$, we have

$$\|\mathcal{P}_S\mathcal{P}_\Omega(\mathcal{T} - \mathcal{A})\|_F$$
$$= \|\mathcal{P}_\Omega(\mathcal{T} - \mathcal{A}) \times_1 U_1^T \times \cdots \times_N U_N^T\|_F \tag{38}$$
$$\leq \frac{N\sqrt{R}}{\lambda}.$$

By (33) and (38), we have

$$\text{RMSE} = \frac{\|\mathcal{D} - \mathcal{A}\|_F}{\sqrt{I^N}}$$
$$\leq \frac{\|\mathcal{E}\|_F}{\sqrt{I^N}} + \frac{\|\mathcal{T} - \mathcal{A}\|_F}{\sqrt{I^N}}$$
$$\leq \frac{\|\mathcal{E}\|_F}{\sqrt{I^N}} + \varphi(\Omega) + \frac{\|\mathcal{P}_S\mathcal{P}_\Omega(\mathcal{T} - \mathcal{A})\|_F}{C_1\sqrt{|\Omega|}}$$
$$\leq \frac{\|\mathcal{E}\|_F}{\sqrt{I^N}} + C\beta \left( \frac{I^{N-1}R\log(I^{N-1})}{|\Omega|} \right)^{\frac{1}{4}} + \frac{N\sqrt{R}}{C_1\lambda\sqrt{|\Omega|}}.$$

This completes the proof. $\square$

**Lower Boundedness of $C_1$**

By the definition of $C_1$, we have that $C_1 \leq 1$. In the following, we will first discuss the lower boundedness of $C_1$, that is, it is lower bounded by a positive constant.

By the characterization of the subdifferentials of norms, we have

$$\partial \|X_0\|_* = \{\eta \mid \langle \eta, X_0 \rangle = \|X_0\|_*, \|\eta\|_2 \leq 1\}. \tag{39}$$

By (35a) and (39), we have

$$\langle \lambda P_n, \mathcal{G}_{(n)} \rangle = \|\mathcal{G}_{(n)}\|_*, \quad \forall n \in \{1, \cdots, N\}.$$

Hence,

$$\left\langle \lambda \sum_{n=1}^{N} \text{refold}(P_n), \mathcal{G} \right\rangle = \sum_{n=1}^{N} \|\mathcal{G}_{(n)}\|_*.$$

Note that $\|A\|_* \geq \|A\|_F$ and $\langle A, B \rangle \leq \|A\|_F \|B\|_F$ for any matrices $A$ and $B$ of the same size. Thus

$$\lambda \left\| \sum_{n=1}^{N} \text{refold}(P_n) \right\|_F \|\mathcal{G}\|_F$$

$$\geq \left\langle \lambda \sum_{n=1}^{N} \text{refold}(P_n), \mathcal{G} \right\rangle = \sum_{n=1}^{N} \|\mathcal{G}_{(n)}\|_*$$

$$\geq N \|\mathcal{G}\|_F.$$

By (35b), $\|\mathcal{G}\|_F > 0$ and $\lambda \neq 0$, thus we obtain

$$\|\mathcal{P}_\Omega(\mathcal{T} - \mathcal{A}) \times_1 U_1^T \times \cdots \times_N U_N^T\|_F$$

$$= \left\| \sum_{n=1}^{N} \text{refold}(P_n) \right\|_F$$

$$\geq \frac{N}{\lambda},$$

i.e.,

$$\|\mathcal{P}_S \mathcal{P}_\Omega(\mathcal{T} - \mathcal{A})\|_F \geq \frac{N}{\lambda}.$$

$\mathcal{G}$ is the optimal solution of the subproblem (32) with the given matrices $U_1, U_2, \cdots, U_N$. By Theorem 1, we have

$$\frac{\lambda}{2} \|\mathcal{P}_\Omega(\mathcal{T} - \mathcal{A})\|_F^2$$

$$< \frac{\lambda}{2} \|\mathcal{P}_\Omega(\mathcal{T} - \mathcal{A})\|_F^2 + \sum_{n=1}^{N} \|\mathcal{G}_{(n)}\|_*$$

$$\leq \frac{\lambda}{2} \|\mathcal{P}_\Omega(\mathcal{T})\|_F^2.$$

Hence,

$$C_1 = \frac{\|\mathcal{P}_S \mathcal{P}_\Omega(\mathcal{T} - \mathcal{A})\|_F}{\|\mathcal{P}_\Omega(\mathcal{T} - \mathcal{A})\|_F}$$

$$\geq \frac{N}{\lambda \|\mathcal{P}_\Omega(\mathcal{T})\|_F}.$$

In other words, $\frac{N}{\lambda \|\mathcal{P}_\Omega(\mathcal{T})\|_F} \leq C_1 \leq 1$. In fact, the value of $C_1$ is usually much greater than its lower bound, as shown in Fig. 4, where the ordinate is the average results on 10 independent random sampling inputs, and the abscissa denotes the sampling rate, which is chosen from $\{0.001, 0.005, 0.01, 0.05, 0.1, \ldots, 0.95, 0.99\}$. Moreover, the regularization parameter $\lambda$ is set to 100. Similarly, we have that $C_2$ in Theorem 5 is lower bounded by a positive constant.

(a) Tensors of size $50 \times 50 \times 50$      (b) Tensors of size $100 \times 100 \times 100$

Figure 4: The average value and standard deviation of $C_1$ vs. sampling rate for gHOI. For a fixed sampling rate, we can see that although $C_1$ is not a constant, the value of $C_1$ is stable with respect to the sampling operator $\mathcal{P}_\Omega$ (Best viewed zoomed in).

## Appendix E: Proof of Theorem 5

*Proof.* Similar to Theorem 4, the following result holds by the first-order optimal condition of the problem (22) with respect to $\mathcal{G}$,

$$\|\mathcal{P}_S \mathscr{X}^*(\mathbf{y} - \mathscr{X}(\mathcal{A}))\|_F \leq \frac{N\sqrt{R}}{\lambda},$$

where $\mathscr{X}^* : \mathbb{R}^m \to \mathbb{R}^{I_1 \times I_2 \cdots \times I_N}$ denotes the adjoint operator of $\mathscr{X}$. By the RSC condition of the linear operator $\mathscr{X}$, we have

$$
\begin{aligned}
\text{RMSE} &= \frac{\|\mathcal{D} - \mathcal{A}\|_F}{\sqrt{I^N}} \\
&\leq \frac{\|\mathscr{X}(\mathcal{D} - \mathcal{A})\|_2}{\sqrt{m\kappa(\mathscr{X})I^N}} \\
&\leq \frac{\|\varepsilon\|_2}{\sqrt{m\kappa(\mathscr{X})I^N}} + \frac{\|\mathbf{y} - \mathscr{X}(\mathcal{A})\|_2}{\sqrt{m\kappa(\mathscr{X})I^N}} \\
&= \frac{\|\varepsilon\|_2}{\sqrt{m\kappa(\mathscr{X})I^N}} + \frac{\|\mathcal{P}_S \mathscr{X}^*(\mathbf{y} - \mathscr{X}(\mathcal{A}))\|_F}{C_2\sqrt{m\kappa(\mathscr{X})I^N}} \\
&\leq \frac{\|\varepsilon\|_2}{\sqrt{m\kappa(\mathscr{X})I^N}} + \frac{N\sqrt{R}}{C_2\lambda\sqrt{m\kappa(\mathscr{X})I^N}}.
\end{aligned}
$$

This completes the proof. □

## Footnotes

[5]http://leitang.net/heterogeneous_network.html