[Reviews · NeurIPS 2014]

Submitted by Assigned_Reviewer_37

The paper introduces an efficient and scalable core tensor Schatten 1-norm minimization (CSNM) method for simultaneous tensor decomposition and completion.
Summary: Section 5 is quite packed.

Submitted by Assigned_Reviewer_41

Summary: this paper presents a method to decompose and complete tensors. The main idea is to represent the tensor as a product of a low-rank core tensor and a set of orthonormal matrices, and then to minimise the nuclear norm of the core tensor (that is, the sum of the nuclear norms of its unfoldings). Intuitively the algorithm mixes fixed rank decomposition (using an upper bound on the rank) with norm-regularisation. The approach is analysed theoretically (in terms of convergence & recovery) and empirically (on one synthetic dataset and one real-world dataset).

Quality: This is a nice idea, and I think it has been executed well. The optimisation is non-trivial and comes with some guarantees, and the scalability of the algorithm is convincing. Results suggest good performance on the chosen dataset compared to a set of state-of-the-art methods.

There are some things that concern me. First, I find the empirical comparison relatively weak as only a single dataset was considered, and this dataset seem to have been used in other tensor factorisation papers (authors, please clarify if this is not the case and the data has indeed been used before). Why not use datasets from other papers, such as the network traffic data of Acar et al, or the Ocean video of Liu et al?

Second, I find the stability guarantees not very convincing. It is true that the middle term in equation 21 diminishes with increasing sample size and hence the impact of the constant C, but without any bounds on C to me this theorem does provide much practical use (that said, Wang & Xu give the same type of guarantee). However, I find the method useful enough to not count this against the paper, it’s an add-on to me.

Question: I am a little surprised that your method is faster than the WTucker method, as my impression is that your essentially solve something like a Tucker decomposition in your inner loop (when estimating your U and G matrices/tensors). Am I missing something?

Clarity: Generally I found this paper to be quite well written. However, I think the description of the algorithm could be slightly improved by making it the currently used set of ranks more explicit. The algorithm is maintaining one rank per mode, which is increased over time (to always reach the R_i, … R_N ranks). If you mention this before equation 9, and that the current ranks are used in this equation (not the R_i chosen in advance), it would be much clearer to me. Second, it would be helpful to discuss briefly implications (and interpretations) of section 4. For example, why could Theorem 4 be helpful even though you have no bound on C?

Originality: I find the main idea relatively original. It brings together a few ideas in a novel way (norm of core tensor = norm of full tensor, ADMM, HOOI, stability analysis a la Wang & Xu). It’s still not gonna change the way of thinking of the field.

Significance: While the method itself isn’t ground breaking, the outcome of increased scalability w/ less sensitivity to rank estimation is quite significant. This could certainly be a method I’d be eager to try (although I’d still like to see more empirical evidence that this works).
Summary: Nice idea that makes sense, combined with good execution in most parts, but theoretical and empirical evaluation could be improved.

Submitted by Assigned_Reviewer_45

The paper presents improved algorithms for tensor decomposition and completion. In settings where the observed tensor has missing values or needs denoising via low rank decompositions, this paper makes the following significant contributions:
a. It shows an equivalence between the nuclear norm of low rank tensor and its core tensor. (I'm not an expert in this area so I dont know if it is new, but I certainly found this interesting and not intuitively obvious)
b. Using the above they formulate a nuclear norm regularized tensor factorization model that scales much better due to the size of the core tensor being much lesser than the original tensor
c. They developed efficient ADMM algorithms for distributed solution of the above problem across several machines
d. They prove rigorous convergence and recovery guarantees theoretically

Experimental results show convincingly that the proposed approach is orders of magnitude faster and also somewhat more accurate than exisiting methods.
Summary: The paper presents improved algorithms for tensor decomposition and completion that is useful in settings where the observed tensor has missing values or needs denoising eg via low rank decompositions

Overall, a very nice paper that addresses an important problem with interesting theory for tensor decomposition, better modeling of the problem, much faster algorithms, and solid theoretical analysis.

Submitted by Assigned_Reviewer_46

This paper introduce a new formulation of tensor decomposition with the purpose of speeding up convex tensor decomposition.

The algorithm is well-designed, and the experimental results show its effectiveness in speed and predictive performance.

Theorem 1 seems quite natural to hold; but it is a good finding and a solid contribution of this paper.

The proposed formulation is not convex in spite of the introduction of the Schatten 1-norms of the unfolding tensors. I would see the variance of the performance of the proposed algorithm.

The datasets are still not quite large; it would be a nice demonstration for the present method to decompose tensors used for recommendation.
Summary: A solid contribution including a new formulation and well-designed algorithm.
Author Feedback
Author rebuttal: We thank the reviewers for their comments and take this opportunity to address their most pressing concerns.

To REVIEWERS 41 and 46:
Reviewers 41 and 46 all commented that more experimental results would strengthen the paper by exploring the scalability and various applications. To address your concern, we will add more experimental results on larger-scale synthetic data (such as 1000*1000*1000), the network traffic data or the Ocean video in the final version. For example, the RSE and the running time, and their standard deviation (std.), of all methods on the Ocean video with 20% sampling rate are listed below, from which it is clear that the proposed CSNM method consistently outperforms the other methods in terms of recovery accuracy and efficiency.

RSE±std, Time(sec)±std:
WTucker: 0.1109±0.0005, 381.46±11.26
WCP: 0.1852±0.0084, 230.48±6.19
FaLRTC: 0.1174±0.0006, 130.82±5.53
Latent: 0.1438±0.0011, 1490.76±20.81
CSNM: 0.1041±0.0002, 31.72±1.43

To REVIEWER 41:
1. First, I find the empirical comparison relatively weak as only a single dataset was considered, and this dataset seem to have been used in other tensor factorization papers (authors, please clarify if this is not the case and the data has indeed been used before). Why not use datasets from other papers, such as the network traffic data of Acar et al, or the Ocean video of Liu et al?

RESPONSE: The BRAINIX dataset is from the OsirX repository, and has been used in [9]. We will add the reference, http://www.osirix-viewer.com/datasets/, to this dataset in our paper. Moreover, we will add more experimental results on larger-scale synthetic data (such as 1000*1000*1000), the network traffic data or the Ocean video. In summary, the proposed CSNM method consistently outperforms the other methods in terms of recovery accuracy and efficiency. Moreover, we will conduct more experiments to evaluate the robustness of the proposed CSNM method to rank estimation in the final version.

2. Second, I find the stability guarantees not very convincing. It is true that the middle term in equation 21 diminishes with increasing sample size and hence the impact of the constant C, but without any bounds on C to me this theorem does provide much practical use. However, I find the method useful enough to not count this against the paper, it’s an add-on to me. Second, it would be helpful to discuss briefly implications (and interpretations) of section 4. For example, why could Theorem 4 be helpful even though you have no bound on C?

RESPONSE: We will discuss the boundedness of both C and C1 in the final version. For example, C can be set to 18+sqrt(2), when |omega|*I^{N-1}*R*log(I^{N-1})>1, and the epsilon is set to 9*beta as in [27]. Please see the proof of Theorem 2 in Appendix A of the Supplementary material for [27]. Moreover, we have added more detailed discussions about Theorem 4 as follows:“Thus, when |omega|>>I^{N-1}*R*log(I^{N-1}), the middle and the last terms of (21) diminish, and the RMSE is essentially bounded by the ‘average’ magnitude of entries of the noise tensor, E. In other words, the proposed method is stable.”

3. I am a little surprised that your method is faster than the WTucker method, as my impression is that your essentially solve something like a Tucker decomposition in your inner loop (when estimating your U and G matrices/tensors).

RESPONSE: In [8], the Polak-Ribiere nonlinear conjugate gradient (NCG) algorithm is used to compute U_n (n=1,…,N) and G for the weighted Tucker decomposition problem. Moreover, the higher-order SVD is first used to initialize the NCG algorithm. As a result, our first-order optimization algorithm is much faster than the NCG algorithm with a line search scheme (note that the line search is time-consuming).

4. Generally I found this paper to be quite well written. However, I think the description of the algorithm could be slightly improved by making it the currently used set of ranks more explicit. The algorithm is maintaining one rank per mode, which is increased over time (to always reach the R_i, … R_N ranks). If you mention this before equation 9, and that the current ranks are used in this equation (not the R_i chosen in advance), it would be much clearer to me.

RESPONSE: Thank you for your comments. We will revise Section 3.2 and improve the description of our algorithm to make the currently used ranks more explicit. We will also revise equation (9) by replacing R_n with r_n, n=1,…, N . Other suggestions will be reflected in the final version of the paper.

To REVIEWER 46:
1. I would see the variance of the performance of the proposed algorithm.

RESPONSE: We will list the variance results of 10 independent runs in Table 1 in the final version. For example, the following results on fourth-order synthetic tensors with 50% sampling rate show that our CSNM method performs significantly better than the other methods in terms of both recovery accuracy and efficiency.

RSE±std, Time(sec)±std:
WTucker: 0.1069±0.0056, 872.56±34.78
WCP: 0.1335±0.0049, 1218.74±26.41
FaLRTC: 0.1452±0.0016, 435.31±19.29
Latent: 0.1180±0.0009, 4031.82±53.82
CSNM: 0.0967±0.0025, 32.29±3.18

2. The datasets are still not quite large; it would be a nice demonstration for the present method to decompose tensors used for recommendation.

RESPONSE: To address your concern, we will add more experimental results on larger-scale synthetic data in Fig. 1 in the final version. For example, the following result on third-order tensor data of size 1000*1000*1000 shows that our CSNM method is orders of magnitude faster than the other methods in handling large dataset.

Time(sec):
WTucker: 120263.49
WCP: 164120.58
FaLRTC: 193813.44
Latent: 335632.96
CSNM: 3266.73

Due to space limitation, we will add a more comprehensive empirical study of tensor data recommendation on three datasets used in [Karatzoglou et al., RecSys 2010] to the appendix of the paper.